# Interconnections: An Analysis of Disassemblable Building Connection Systems towards a Circular Economy

**Timothy M. O'Grady** [1,*] **, Roberto Minunno** [1] **, Heap-Yih Chong** [2] **and Greg M. Morrison** [1]

1   Curtin University Sustainability Policy (CUSP) Institute, Curtin University, Bentley, WA 6102, Australia;
    roberto.minunno@curtin.edu.au (R.M.); greg.morrison@curtin.edu.au (G.M.M.)
2   School of Design and the Built Environment, Curtin University, Kent Street, Bentley, WA 6102, Australia;
    heap-yih.chong@curtin.edu.au
*   Correspondence: timothy.ogrady@postgrad.curtin.edu.au; Tel.: +61-432-733-246

**Abstract:** This study investigates the interconnection methods used to create a circular economy building featuring modularity and designed for disassembly and relocation. Designing modular buildings for disassembly and reuse can decrease waste production and material depletion, in line with the circular economy framework. Disassemblable buildings require connections to be easily accessible. Visible connections may be unpopular features; however, concealing these, yet leaving these accessible, presents a substantial design challenge. This study demonstrates solutions to this challenge by analyzing a purposely designed case study: the Legacy Living Lab. The challenges of disguising and sealing, such as by waterproofing, two types of connections are analysed: structural and non-structural. This study details the materials and connections used across the two analyzed connection types and compares the weights and reusability of components. Thus, a necessary case study is provided for practitioners to advance circular economy theory in the building industry. Notably, all connections in the Legacy Living Lab can be easily accessed with standard building tools, facilitating its disassembly and fostering component reusability.

**Keywords:** circular economy; prefabricated construction; design; disassembly; deconstruction; resilience

## 1. Introduction

The construction industry generates the largest percentage of waste globally [1,2]. Buildings produce a more significant proportion of this waste when demolished at the end of life [3]. Over the past 50 years, scholars have studied construction and demolition waste and have been concerned with the sizeable units and large volumes of waste components [4]. A possible solution to the size and weight of building components is reducing them into smaller parts through design for disassembly [5]. Design for disassembly may also enable the reuse of the deconstructed components, saving materials from landfills and avoiding additional resource consumption through producing new components [6].

The literature that has attempted to solve the issue of the generation of large volumes of construction and demolition waste focuses typically on material recycling [7]. Material recycling can be an advantageous strategy, but in the case of construction materials, the preferred solution is introducing these as replacement feedstock to remanufacture components, a practice referred to as downcycling [8–10]. To prevent the environmental effects of landfills and downcycling, scholars and practitioners should consider end-of-life constructions as valuable resources rather than waste, as suggested by the circular economy approach [10–12]. A circular economy in construction can be defined as the economic system that replaces the end-of-life concept with reducing, alternatively reusing, recycling and recovering materials in the production/distribution and consumption process [13]. Hence, in a circular economy approach, the reuse of components would be enabled primarily by design for disassembly [14]. This approach would allow the rescue and restoration of

components at the end of buildings' service life, thereby limiting demolition waste to a minimum [15,16].

A few recent case studies have demonstrated the environmental benefits of reusing building components [17,18]. Further, technological innovation fosters the disassemblability of structural components, such as concrete columns, floor systems and roof structures [18–21]. However, many barriers hinder the reuse of building components. For example, demolishing and recycling are often more financially feasible than disassembly [22], whereas deconstructing remains complex and inconvenient. In addition, in many countries, the lack of a market for reusable components represents a substantial barrier to establishing a closed-loop supply chain in the construction industry [23,24].

Owing to its many benefits, design for disassembly and reuse is gaining traction [22,25–27], although the limited knowledge on reusing building components presents a noticeable literature gap [28]. Despite the many initiatives to unlock the potential reuse of building materials, empirical studies on the topic are lacking [13,28].

To fill this literature gap, this study explores a category of building technology that could represent a straightforward and close-to-market solution for disassembly and reuse: prefabricated buildings. Due to their nature, prefabricated buildings may solve some of the challenges of reusing building components [8]. For example, one challenge is that buildings are designed and built to satisfy spatial boundaries, and, consequently, each building has substantially different dimensions. For prefabrication, similar dimensions and mechanical characteristics would facilitate closing the material loop, allowing the integration of old components into new construction [8,29–31]. One subcategory of prefabricated construction is volumetric modular buildings, created using boxlike structures built offsite, to be transported and assembled onsite [32]. These buildings are widely used in many northern European countries and the United States [33,34]. However, in other countries, including Australia, their adoption is hindered by an overall negative perception. In these countries, volumetric modular construction has been used in mining camps, emergency accommodation and social housing projects, which has created a customer perception that these are temporary, inelegant buildings [35]. Therefore, high quality prefabricated buildings should be produced, which would improve public opinion about the brand quality of modular buildings. Such high quality buildings could begin to shift the public's perception of prefabricated buildings through new anecdotal learnings [35].

Against this backdrop, this study addresses the following research questions:

- How can building components be designed according to the circular economy framework?
- How can buildings be assembled so that their connections are accessible and disassemblable?
- How can connections be accessible without affecting the visual presentation of buildings?

Addressing these research questions is important because the construction sector is slowly embracing the circular economy framework [5,13]. As many contemporary researchers assert (see, for example, [5,8,23,26]), modular buildings could be designed for disassembly and reuse, however, doing so might result in accessible and visible joints [29], which could negatively affect the visual presentation of buildings. Buildings where the connections are visible are labelled by customers as low quality buildings [35], while this paper promotes the concept that modular building could be both disassemblable and high quality. In exploring possible solutions to these questions, the case study method investigates multiple connection solutions adopted in a fully disassemblable modular building, namely, the Legacy Living Lab (L3). This study describes the magnetic, bolted, screwed and flashing connection details adopted in L3, revealing their strengths and weaknesses. Further, it aims to guide practitioners on ways to design a building for disassembly effectively.

## 2. Materials and Methods

### 2.1. Case Study Method

Three main reasons account for the choice of the case study approach to provide insights on ways to design and manufacture accessible yet concealed connections. First, the

case study method includes a thorough explanation of how L3 was designed and built as an eight module disassemblable circular economy building in which all the connections are concealed. Second, through this approach, the barriers to, and solutions for, manufacturing disassemblable connections are revealed. Third, the L3 case study may support the upscaling of the technologies used and provide insights on the adoption of the same connections in similar projects. Moreover, the case study method is considered the best tool to observe and research a practical phenomenon, and is viewed as an empirical endeavour [36,37].

Further, the related literature has called for practical applications to progress the construction industry towards design for disassembly [5]. Indeed, although many theoretical examples are available of buildings and components designed for disassembly and reuse, more research is required to integrate theory with practice [13]. For these reasons, this study applies the case study method through L3, designed and built to prove the concept of disassembly and reuse. Therefore, a thorough analysis of L3 can demonstrate the strengths and weaknesses of multiple connections of a building designed for disassembly—L3. This method unfolds in four main steps, which were adapted from Tellis [38], as follows:

*(1) Case study design and construction:* In this step, the L3 case study was designed as a building representing an extreme case of disassemblable buildings (i.e., the typical modus operandi is not used—L3 is described in detail in Section 2.3). A research team member selected the most appropriate connections to evaluate in this study, using his professional knowledge concerning prefabricated buildings.

*(2) Design details:* The selected design details were collated and reported according to their assembly process with photographs and videos. After connections were collated, these were categorised according to their material type, weight and reusability of the materials were documented.

*(3) Sample analysis:* After the building was constructed, evidence of the concealed connections' technical and physical characteristics was analysed. The L3 research team applied several methods to conceal the connections while leaving them accessible. Some connections were easy to access, but others required tools to be used for access. Another distinctive characteristic of accessible connections is their feasibility for disconnection. For example, the connections of load-bearing components depend on the load they bear since this load must be removed before disassembly. Other characteristics are the air and water tightness of the connection.

*(4) Reporting:* In this step, recommendations and implications concerning adopting different connections were reported. This step is crucial because it focuses on best practice alternatives when designing and assembling a disassemblable building, based on this successful case study.

### 2.2. Theoretical Framework

This study explored the intersection between environmental strategies to decrease construction and demolition waste and technical aspects to create an aesthetically pleasing building with concealed connections. Figure 1 presents this study's theoretical and empirical framework.

This theoretical framework adopts elements from three fields: a design for environment, architectural design and construction technologies. The circular economy, the elements of the 3Rs framework (reduce, reuse, and recycle; explained in detail in Sections 2.2.1–2.2.3), and material saving are the key concepts originating from the research field of design for the environment [10]. Specifically, the 3Rs framework can be considered a procedural hierarchy in which reduce is the most effective practice to decrease environmental impact, followed by reuse and recycling [19,21,26]. Similarly, the construction technology investigated is building disassemblability, fostered by accessible connections, and finds optimal application in modular or prefabricated buildings (e.g., see the connections investigated by Derikvand and Fink [39]). Therefore, these two aspects (environmental and technical) provide the basis to deliver a high quality building that considers architectural design to incorporate desirable finishes made possible by carefully concealed connections.

Specifically, the 3Rs concept establishes the reduction of materials, the reuse of components and the recycling of the waste in a building. In adopting this theoretical framework, this paper delivers practical solutions for builders, who are urged to apply circular economy concepts yet are concerned with the customer response to the architectural features of their buildings.

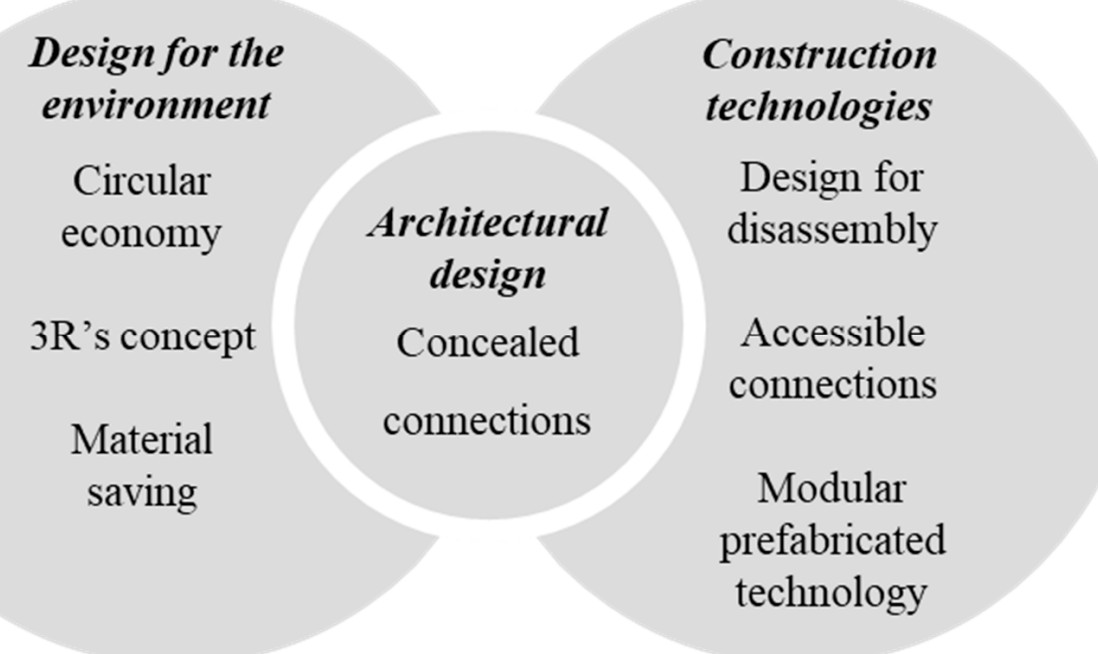

**Figure 1.** The theoretical framework of this study.

### 2.2.1. Reduction of Materials

Scholars and practitioners have proposed strategies to reduce the use of resources and materials in the construction sector. The most prominent strategy is to optimise material use by reducing waste production and using only essential resources [40]. Lean construction methods efficiently reduce offcuts and, subsequently, waste [41], whereas accurate engineering reduces redundant and overused resources [21].

Another approach towards reducing material use is ensuring product life extension. Concerning buildings, designers can achieve life extension by increasing the buildings' material quality and resilience by creating flexible spaces [8]. Material resilience is described as a structure that must be robust enough to withstand the forces imposed on it through the disassembly and relocation process [42]. In this context, building relocation can also enable material reduction, preventing both demolition waste and the use of materials to reconstruct a building elsewhere [18].

### 2.2.2. Reuse of Building Components

Building material reuse implies the separation of components from each other. Designing for disassembly facilitates component and material separation. For materials designed to be removed, permanent connections, such as welded or glued connections, must be avoided [43]. This aspect is often overlooked, since disassemblable connections must be accessible, allowing technicians to quickly and safely remove components [21]. Thorough planning of disassembly procedures is required to enable the safe, timely completion of the operation. In this process, the weight of individual components and the structural load transferred through components must be considered.

### 2.2.3. Recycling of Construction and Demolition Waste

Recycling involves transforming materials through mechanical, thermal or chemical processes [44]. Although it saves substantial resources in most cases, it is at the bottom of the 3Rs hierarchy because it also requires consuming additional materials and fuels [8,45]. Further, building materials are typically downcycled rather than recycled, since they are transformed into products of lesser value [46]. For instance, concrete must be cleared from steel reinforcement bars and other contaminants and then crushed to form gravel. To turn gravel into new concrete requires additional cement, water and sand [47].

### 2.3. Case Study Description and Study Boundaries

The Legacy Living Lab (L3, Figure 2) connections are analysed as a case study in this paper. L3 is an eight module prefabricated building that the research team designed and built to demonstrate the circular economy's applicability through the disassembly and reuse of building components. L3 was built as a joint effort between the research team, who lead the conceptualisation and project management, as well as supervised the construction process, whilst the construction and detailed engineering were carried out by an Australian modular building company at a dedicated prefabricated manufacturing facility in Perth, Western Australia. L3 has a floor area of 251 m$^2$ divided into two floors and includes a commercial area and café space on the ground floor and an open office area with a shared kitchenette on the first floor. The ceiling and internal cladding were designed to be fully disassemblable, thus granting access to the insulation and the mechanical and electrical plant systems without creating any waste (Figure 2). Both floors of the prototype L3 are designed for adaptability.

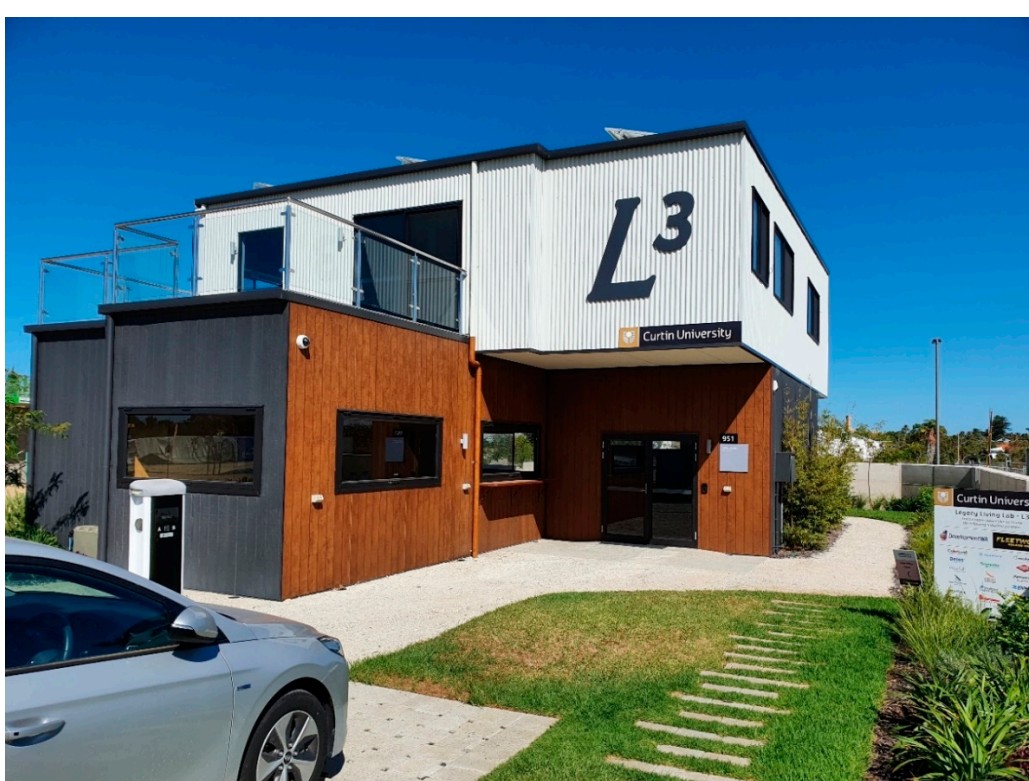

**Figure 2.** The case study selected for this research, the Legacy Living Lab.

The design features large open spaces that can be closed for future use and redundant features such as service ducts and elevator reveal for future adaptability. The structure is built using heavy gauge steel chassis and columns, light gauge steel wall frames, and particleboard floors. The external cladding is pressed timber on the ground floor and corrugated steel sheet on the first floor and roof. The internal cladding is composed

of plywood sheets, plasterboard and acoustic ceiling panels. Although many types of connections are used within L3, this study's boundaries are restricted to the structural interconnections (i.e., the connections between the eight modules and between the modules and foundations), interior connections and flashing connection systems.

## 3. Results

A total mass of 36.3 tonnes was used in L3, including steel (61%) and timber (26%), of which 58% is disassemblable and technically reusable infinite times due to the resiliency of that material and the accurate design for disassembly (see Figure 3). Conversely, the reuse of materials such as timber is assumed to be limited to three times, because of material degradation and the use of potentially toxic chemical treatments such as termite proofing [42]. This section is organised into five subsections based on material categorisation: steel, timber, cladding and floor covering, insulation and glazing, and waterproofing and plumbing details.

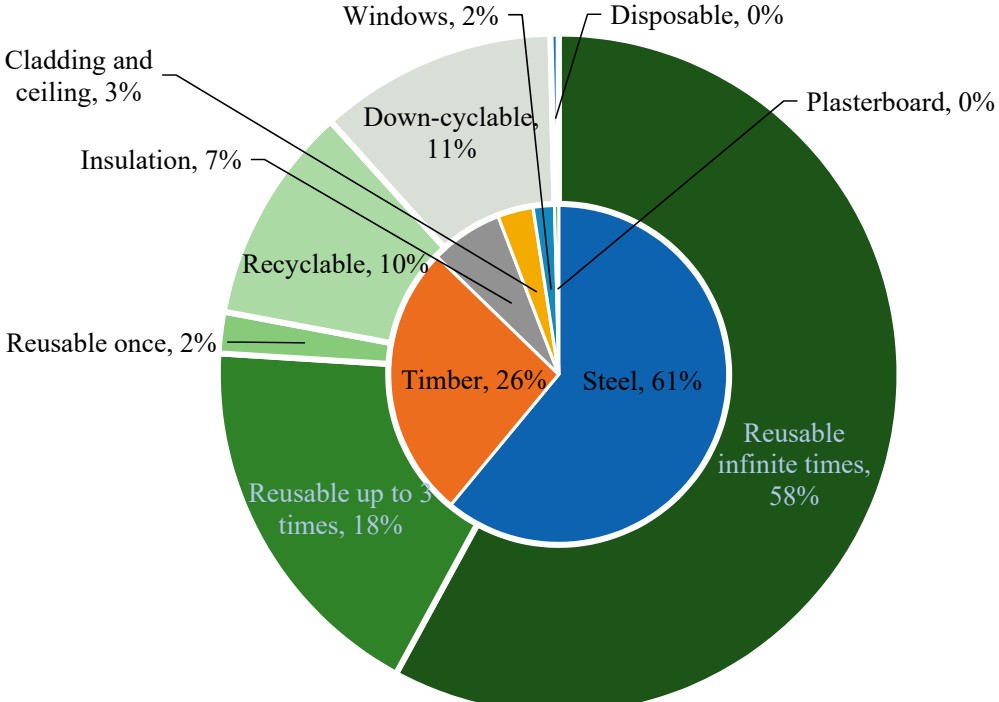

**Figure 3.** Pie and doughnut chart of material and subsequent product use of materials used in the Legacy Living Lab.

Throughout the design of L3, every effort was made to ensure that there were no visible joints between each of the eight modules. The goal was to create a building that would not emphasise its modularity and to use, wherever possible, connection methods prescribed in traditional construction practice to ensure the prefabricated construction typology was not prominent. This section discusses the construction details to reveal the effort made to conceal the disassemblable joints to maintain the aesthetics of the building and conceal the construction methodology used. The design modus operandi was simple, and the following criteria needed to be met to reframe the public perception of prefabricated construction in Australia [48]:

- It should be possible to dismantle the building into eight modules so that it can be moved;
- Waste created from removal or renovation must be eliminated or reduced;
- Connection details had to be engineered and practical to implement for a full scale building.

Following these design criteria, this study investigates the interconnections necessary to facilitate a circular economy building design. Section 3 shows how connection details,

such as balcony waterproofing, flashing design and material connection details, were used in this circular economy building. L3 is a full size building, a constructed building that meets the Building Code of Australia for commercial construction [49]. The following sections discuss the internal and external finishes and the structural and services connections that facilitate the circular economy of the L3 case study.

*3.1. Steel*

3.1.1. Structural Steel Frame—Bolted Connections

The steel frames used in the L3 prototype were removed from the recycling stream. These eight structural frame modules were salvaged after they were produced for a third party builder who went bankrupt. After the contract was terminated, there were 42 structural frames with no use or commercial value. Steel is a highly recoverable material that, if properly maintained, can be recycled with no loss in structural properties [6]. These frames were intended to be cut down and recycled. However, the carbon intense process of recycling provided the insight that the resource should be reused as it is, with recycling being the final option for the material [18]. The research team was able to design the building to incorporate the structures with minimal redesign work. This allowed 18 tonnes of the existing structural steel to be reused (Table 1). Once the configuration layout was completed, the steel structures were sandblasted and were re-engineered to ensure the long span overhangs would be selfsupporting.

**Table 1.** Steel components used in the L3 case study by mass (kg) and their resilience metric. In = reusable infinite times, Rc = recyclable.

| Component Description | Material Resilience | Mass (kg) |
|---|---|---|
| Steel chassis and load-bearing structure | (In) | 16,138.9 |
| Stairway steel structure | (Rc) | 422.8 |
| Lightweight steel structure; internal walls | (In) | 3590.3 |
| Steel sheets used on the first floor for external cladding | (In) | 652.1 |
| Steel sheets used for roof covering | (In) | 531.3 |
| Bolts and nuts | (Rc) | 97.7 |
| Screw pile lightweight steel foundations | (Rc) | 735 |
| **Total weight** | | **22,168.10** |

The structural steel frame connections were bolted together behind the various detailed internal finishes to ensure no visible connections. For the bolted interconnections, 68 × No 16 mm bolts were used. Traditionally, prefabricated buildings are taken to a site and welded into position, creating a permanent fixture. Bolting each of the module connections enables the building to be disassembled without creating waste from a chemically bonded or welded connection, unlike concrete or traditional steel structures.

3.1.2. Internal Balustrading

The internal balustrade and handrail design was an intricate detail to solve. As L3 is prefabricated, the staircase spans two levels, containing vertical module connections delivered to the site separately. A novel design was developed to maximise the level of work completed off site and facilitate the easy disconnection and transportation of the complete building. The balustrade was split into two sections, each contained within a separate module. Figure 4 shows the dot-matrix model scan from which the balustrade was modelled, which shows the module disconnection points.

On the first floor, the staircase edge-fall protection is maintained. The balustrade follows the staircase opening around the first floor with no interconnections between the first and ground floor modules. The handrail follows the staircase internally on the far wall to give continuous support when transitioning from one level to the next. To increase its disassemblability, the handrail is fixed to the wall with screws, enabling easy

removal between modules for transportation. The design features of the internal balustrade system promoted the circular economy concept through the reuse of components and was facilitated by bolted connections in place of welded connections between the rail and the wall brackets, handrail angles and infill panelling.

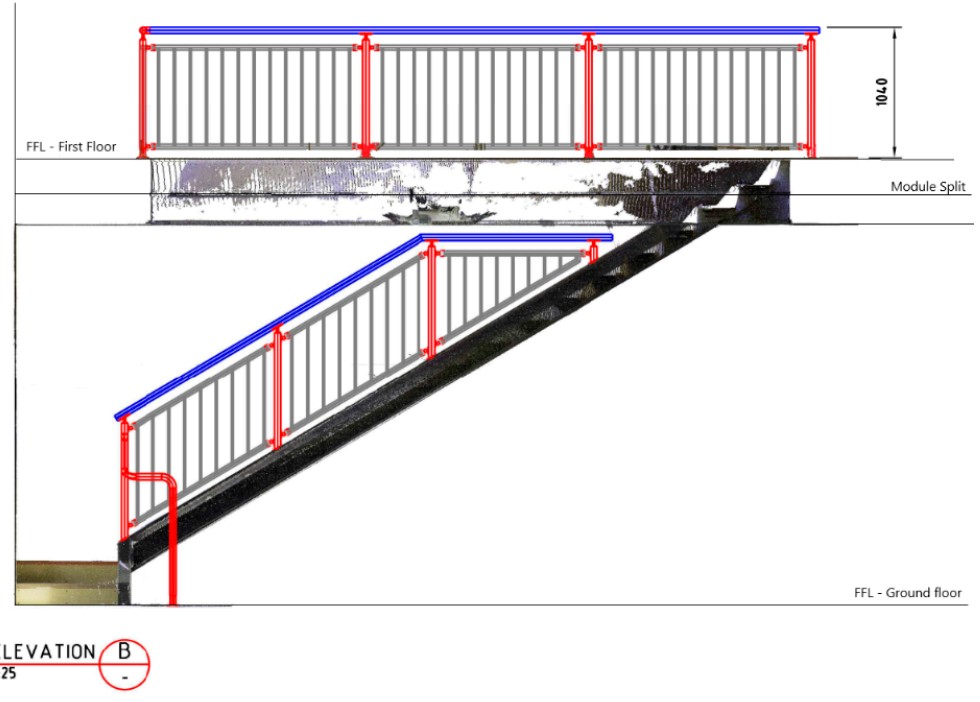

**Figure 4.** Balustrade design scan. The balustrade design fosters disassembly; the balustrade runs under the ceiling on the ground module and is selfcontained within the first floor module.

### 3.1.3. External Roof and Wall Cladding—First Floor

Two types of cladding were used to cover the building's exterior—the use of two different materials assisted in concealing an interconnection between the ground- and first-floor modules. Using a flashing joint between two different cladding materials is standard practice. However, using the same material and joining between the ground and first floor modules creates visual evidence that the building construction method differs from standard construction methods. It is posited that these minor incremental design improvements will prevent users from quickly identifying that they are in a prefabricated structure and will remove the current negative perceptions about such structures.

The roof and wall cladding sheet layout was also designed to enable overlaps of corrugated sheeting to meet over module interconnections to foster easy disassembly. The result of using this layout is that a full sheet need not be removed for deconstruction and can remain in place with one screw line to support the sheet's weight. The screws can be installed for transport once the module is craned off the building. Adopting this method reduces the extent of work performed at a height and the need to lift and remove a sheet whilst in an elevated working position. The layout of air conditioning compressors and solar panel arrays were also considered in designing the roof sheet spacing. This design allowed each of the services installed on the roof to be clear of the lap sheets, enabling improved disassembly. Corrugated steel sheets were selected for their high resiliency and ability to be recycled at the end of life. A screwed fixing method enables the sheets to be easily replaced throughout the deconstruction. It promotes the separation of materials at the end of life, allowing the material to continue in the loop.

### 3.1.4. Footing System

The traditional footing system specified for a building of this size would be a concrete pad footing with concrete doughnuts to maintain the required air gap between the ground

and building base. The engineers' drawings showed that two concrete sizes were employed (0.38 and 0.64 m$^3$) (Figure 5a,b).

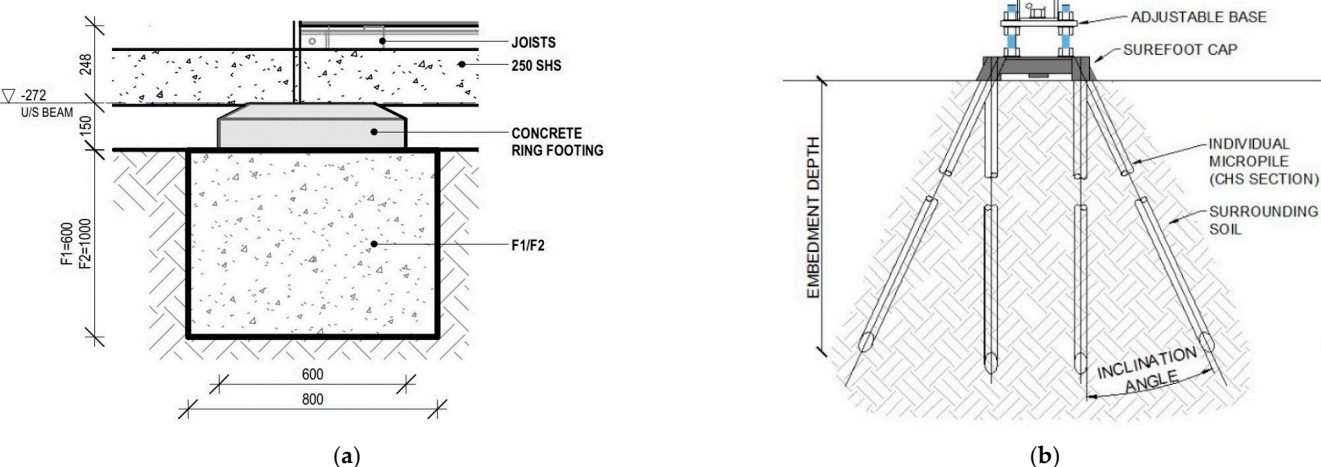

**Figure 5.** (**a**) Traditional footing design; mass concrete footing with a concrete ring to separate the building; (**b**) concrete free micropile system used as the foundation of L3.

The practicality of concrete footing systems is limited in the longer term, given that concrete can only be downcycled rather than recycled [47]. There are opportunities to reuse concrete footings; however, the associated cranage and transport of engineered footings negate any environmental benefit in specifying them in the first instance [50]. A new footing technology was used to reduce the use of concrete in the building, which incorporated embedding a series of steel micropile tubes through a load-bearing baseplate. This base plate is a welded galvanized steel plate with guide tubes welded to its external edges at predetermined angles. The micropile tubes were mechanically embedded into the foundation at predetermined lengths prescribed by an engineer. The connection details in these all-steel footing systems enable the complete assembly to be disconnected at the end of life through a bolted connection. The use of this method eliminated the need to excavate trenches and pour concrete, as well as the curing time. Instead, micropile footings are ready immediately after installation. From an environmental impact point of view, the micropile system allows saving of materials as well as decreasing the greenhouse gas emissions related to footings. In the L3 case, the amount of concrete specified would have been 19.85 tonnes, while only 735 kg of steel were employed for the alternative system, representing a material saving of 96%. In terms of carbon emissions, that translates into 3772 kg $CO_2$ eq (19,850 kg × 0.19 kg $CO_2$ eq/kg) for the concrete option, versus 1176 kg $CO_2$ eq (735 kg × 1.6 kg $CO_2$ eq/kg) for the micropile alternative, representing a saving by 69% in greenhouse gas emissions. The carbon emission coefficients for concrete and steel (0.19 and 1.6 kg $CO_2$ eq/kg), were measured in kilograms of carbon dioxide equivalent and sourced from [6].

### 3.2. Timber

### 3.2.1. Internal Wall Lining

Wall linings provide the finished surface that will define the internal space of the building. Traditionally, buildings have a plastered internal finish, whether the building is timber framed, where drywall is often used, or wet set plaster with a float finish, as is often the case for brick buildings. The specified construction typology is important when designing for a circular economy building because most of the services will travel within the wall cavities and roof spaces throughout the building. It is important to provide accessible walls and roof cavities. The internal wall finishes of both construction types (timber frame and double brick) limit the ability to install or change internal services without creating waste. To apply the circular economy approach to an area of the building

where disconnection, disassembly and access are extremely limited, a disassemblable wall lining detail was designed, which was resilient enough to be removed on multiple occasions without creating waste.

To circumvent the issues in the common construction process, a plywood wall lining system was designed in which each panel was individually fixed, which facilitates the easy removal of the internal wall lining to access wall cavities. Access to internal wall cavities enables insulation material and plumbing and electrical layouts to be easily modified throughout the building's life cycle, which extends its life. The plywood system facilitates developing products in the future because it incorporates the industry standard sizing of 2400 × 1200 mm (where possible). A clear urethane coating was applied to both sides of the plywood board to accentuate the natural grain finish of the internal surface and to reduce the possibility of moisture absorption from within the external wall cavity. The plywood was located within an aluminium negative detail, which was designed to assist in positioning the panels during removal and reassembly, by providing a framed working edge. Each panel was fixed with exposed screws, creating a noncontinuous surface covering. Table 2 lists the timber components used in L3.

**Table 2.** Timber components used in the L3 case study and their resilience metric. In = reusable infinite times, 3t = reusable up to three times, and Dc = downcyclable.

| Component Description | Material Resilience | Mass [kg] |
|---|---|---|
| Pressed fibre particle board used as floor structure | (Dc) | 4102.6 |
| Plywood covering internal walls and the first floor ceiling | (3t) | 3328.0 |
| Pressed timber used as ground floor external cladding | (3t) | 1811.7 |
| Reused timber stair treads | (3t) | 164.0 |
| Internal timber doors | (In) | 138.0 |
| **Total weight** | | **9544.30** |

This plywood wall lining system allowed the panels to cover the module interconnections. Covering these interconnections with a plywood panel that spans the connection diverts occupants' eyes from the true modularity of the building. The panels also allow easy access to facilitate the removal of structural bolted connections. Traditionally, volumetric modular prefabricated buildings would have a moulding to cover the intersection of a structural connection between modules. This moulding would be used to finish the connection between modules; however, this creates a finish that differs from what would be expected in a traditional building. To accompany the moulding between modules, traditional prefabricated buildings use ceiling access panels to allow access to structural connections within the roof spaces. However, the association with module joints and access panels will lead to a negative interpretation among owners and occupants [35]. The layout of the plywood lining system solves the problem of having unsightly access panels by creating an entire finish that consists of larger plywood access panels. The design considers the material that must be removed when transporting the building. Figure 6 shows the module joints. It reveals that a minimal number of panels need to be removed during the transportation process and that covering the intersecting module connections improves the aesthetics of the building's interior.

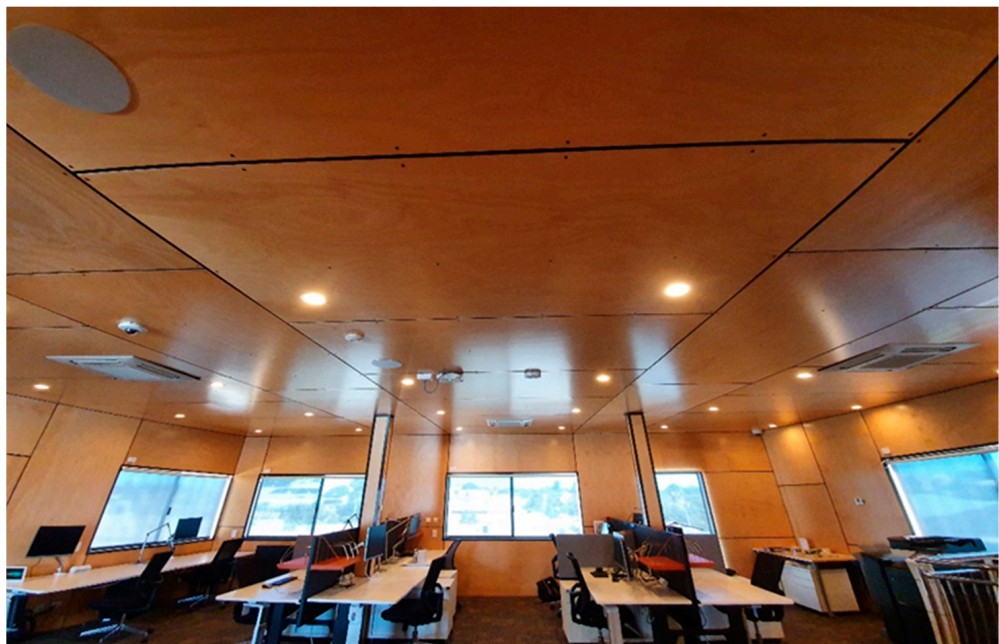

**Figure 6.** First floor of the prototype building L3. The wall and roof cladding is disassemblable, showing the frames granting access to insulation and systems. This feature allows low waste production and low environmental impact during the L3 operation stage.

### 3.2.2. External Wall Cladding—Ground Floor

Wall cladding can be fixed using different methods, depending on the selected material. The pressed timber selected for L3 was chosen because it is produced using natural products (i.e., a mix of pulped timber and wax) that are heated and pressed into a solid board. This board resembles timber and performs in a similar way as timber. Further, it can be used instead of the cement based wall panelling products that are typically used in this application.

To maintain a traditional building aesthetic, the external module joints were designed to meet at 90 degree internal and external angles so that standard aluminum moldings could be used. The aluminum profiles that join the pressed timber sheets were color coded for a natural appearance. Screws were used to fix the timber panels into position—and not the spiral nails traditionally used—for two reasons. First, the panels had to be made disassemblable for future adaption and reuse. Second, it had to be ensured that the building could withstand the loads not usually imposed on a commercial property, considering that it would have to withstand wind speeds of up to 110 km/h during transportation.

### 3.3. Cladding and Floor Covering

### 3.3.1. Ceiling Cladding

Noise cancelling acoustic felt cladding was used for the ground floor and the apartment display space ceiling because the increased ceiling height and the collaborative meeting space, as well as the breakout apartment space that may be a venue for discussions, would require an acoustic treatment to reduce ambient sound and eliminate footfall transmissions from the first floor. To promote disassembly further, a new fixing method was prototyped, with the ceiling being held in place using magnets. The 2400 × 1200 mm panels each had three steel tracks attached to their internal face. Each of these steel tracks corresponded with 4 × No. 15 kN rare earth magnets. The combined magnetic force over a panel was 180 kg, and the panel weight was just 4.9 kg (Table 3).

**Table 3.** Cladding and floor covering materials used in the L3 case study by mass (kg) and their resilience metric. 3t = reusable up to three times, 1t = reusable once, and D = disposable.

| Component Description | Material Resilience | Mass [kg] |
| --- | --- | --- |
| Vinyl for covering floors in wet areas | (1t) | 714 |
| Magnetic felt ceiling | (3t) | 333.7 |
| Carpet covering 193 m$^2$ of internal floors | (3t) | 183.4 |
| Plasterboard cladding, for kitchen/bathroom areas and ground-floor ceiling | (D) | 140.2 |
| **Total weight** | | **1371.30** |

The circular economy based design thinking behind the acoustic ceiling panels was to have a highly efficient surface, which comprised the internal lining and the finish for the first floor. These panels can be easily removed if retrofit is needed, or for transportation, and eliminate the need to use a molding or ceiling panel for access. The success of the design was demonstrated with the installation of the security system into L3. The university card access and credential system had to be installed into the building after the completion of its internal finishes, during which the entire ceiling system was removed in a matter of minutes. The ease of removal and reinstallation demonstrated the flexibility of L3, adding to the life extension of the building through retrofit and adaption. The panels also contain a high content of recycled PET plastic bottles, which promotes the circular economy by specifying manufacturers using nonvirgin feedstock for their production.

### 3.3.2. Carpet Tiles

Paints and floor covering products are identified as important sources of volatile organic compounds (VOCs) in the indoor environment, with carpet adhesives providing the second largest emission rate of VOCs after wall and floor construction adhesives [51,52]. The production of carpet tiles in the European Union alone represents 100 million kg per annum of a largely nonrenewable resource [53]. Typically, carpet tiles are glued into position on a building's substrate, creating a durable wearing finish for high traffic areas, such as office spaces. Advances have been made in the adhesives used in carpet tiles to facilitate a more efficient recycling process, with switchable adhesives being trialed to develop carpet tiles into a closed loop product [53]. The drawbacks of the carpet tile can be weighed against its advantages, because having a modular system can minimize the waste generated throughout the operational life of a building. The modular system enables individual tiles to be replaced when worn or damaged, thus enabling the extension of life for the remainder of the carpet. Traditional rolled carpets might require the entire carpet to be lifted in order to make good a small damaged or soiled area of the carpeted surface of a room.

To reduce the environmental impact of the carpet tiles, two initiatives were implemented to promote the reduction in chemical connections, increase life extension and facilitate the reuse of carpet tiles in L3. First, the carpet tiles selected for L3 were in their second life cycle because the supplier had recovered these from another building. Although the tiles were in perfect condition, they were removed upon being deemed to be outdated when the building was leased to new tenants. Second, the fixing method selected was a double sided tactile pad, which meant that wet application glue was not used to fix the flooring to the building's substrate. Instead, the carpet tiles were effectively fixed to each other, acting as a single floating floor, rather than each individual tile being glued to the substrate. This method revealed an additional advantage—there was reduced off gassing of VOCs during the first year of the building operation. The tiles can easily be replaced, since it is far easier to remove individual tiles fixed without using heavy adhesives. The drawback of this fixing method is that the carpet tiles must be stored and kept flat during transportation and storage since the fixing pad has only enough adhesion to keep the tiles fixed when flat, whereas typical flooring contact adhesives can fix some bending of the

tiles. Nevertheless, using this method enables the complete recovery of the carpet tiles for use in a second life cycle, with the ability to continue in the loop. Last, since adhesives are not used, the carpet tiles can be chipped down and reintroduced as feedstock during the recycling process for new tiles at end of life, completing the hierarchy of the three Rs.

*3.4. Insulation and Glazing*

3.4.1. Windows

In most construction typologies, windows can be easily disassembled and reused [17,54,55]. In Western Australia, where the building in this case study was constructed, it is a slightly different case because the double brick construction limits the ease with which windows can be removed from most buildings. To have a greater impact on the local industry, window innovation was needed to consider the possibility of retrofitting new windows into existing frames.

This consideration led to the development and implementation of two new products. The first solved the issue of an operable window that would enable the inclusion of a serving space at the café space. The second featured a window system that would incorporate an interchangeable glazing panel that could be tested and scaled to market. To keep in theme, the windows were kept (as much as possible) at the standard industry sheet size of 2400 × 1200 mm to allow retrofit and reduce waste [8,30,31].

The operable windows in the café space have been developed out of a cost engineering solution for a bifold serving window (typically, four window panels on a common track). To overcome the cost of the bifold windows, a novel idea was used—the gas struts were attached to gas struts to a single commercial door, in order to have a hinged awning window that opened from the bottom up. This design offered the multiple benefits of a larger viewing area through the window and better protection from weather in light weather events, since the window had an awning and was not just an exposed opening.

Windows were used to conceal redundant features designed into the building to promote its life extension. Designing in redundant features enables the building to be adapted later in its operational life, limiting the waste created in the process. Windows were installed in lieu of future elevator reveals. These single pane windows act as functioning pre-cut holes in the walls to allow elevator or ramp access in a future use case. Table 4 lists the mass of the windows.

**Table 4.** Insulation and glazing materials used in the L3 case study by mass (kg) and their resilience metric. 3t = reusable up to three times, and Rc = recyclable.

| Component Description | Material Resilience | Mass [kg] |
|---|---|---|
| Aluminium, stainless and glazing | (3t) | 744 |
| Insulation | (Rc) | 2516 |
| **Total weight** | | **36,343.70** |

3.4.2. External Balustrades

The external balustrade was designed for disassembly so that the building may be lifted from its location and transported. Transport height is a major consideration in prefabricated design, considering that the building cannot be easily moved through central road corridors if the height of the load exceeds 4.3 m [56]. Thus, the external glass balustrading system was designed to be removed for transportation. To achieve this, threaded studs were installed, which were embedded into the balcony floor through to the structural steel. This process enables the connection between the posts and the structural steel module to not be hard-wearing, as is typical with a screwed connection that is subject to the lateral forces that can be expected in balustrading. Since the posts can be removed easily, the glass panels between them were fixed using grub screws and washers, and the panels can be laid flat within the building when removed. The handrail needed to be removable, and, due to its size, needed to be broken down into pieces, in order to be

stored internally for transportation. Given that the common practice is to use tungsten inert gas welding for external connections, a disassemblable connection detail was needed that would be strong enough to handle the forces and maintain a reasonable aesthetic. The solution was to drill and tap a grub screw that would be located beneath the handrail that would be concealed and allow the disconnection of the handrail at each 90-degree angle joint.

### 3.5. Waterproofing and Plumbing
3.5.1. Balcony Waterproofing

The deconstruction and transportation of a building is difficult without creating additional waste, and the use of new resources should be minimized at all costs. For this reason, designing a disassemblable balcony system was particularly important in this project. One requirement of the L3 case study was to incorporate a balcony to give potential precinct buyers the aspect of the view from their first floor. The balcony presented waterproofing issues because the plan and section view module connections both needed to be considered in this one section of watertight flooring. The standard practice in this scenario is to waterproof the balcony substrate with a trafficable membrane once the modules are assembled onsite, which creates a continuous membrane that is cut when the building is deconstructed. This practice not only creates additional work and waste, but also builds up the surface level of the balcony each time the membrane is cut and then a patched section is applied over the module joint. Therefore, for L3, several design iterations were worked through until the final design: a silo water tank style flashing method was combined with an overflashing, and these were installed. This flashing ran down the balcony gradient (Figure 7a), providing a lateral connection between the ground floor modules (Figure 7b). An overflashing was used to vertically connect the intersection of the top of the ground floor modules and the bottom of the first floor modules (Figure 7b). To conceal the waterproofing method, a flat trafficable surface was provided and to maintain as much as possible the illusion that the L3 was a traditional building and not prefabricated, a modular, permeable paving system was selected to cover the details (Figure 7c,d). The balcony was constructed at a 1:90 degree fall away from the first floor of the building, allowing rainwater to travel through the paving system, onto a waterproof trafficable membrane, to be collected by a gutter and captured as stormwater.

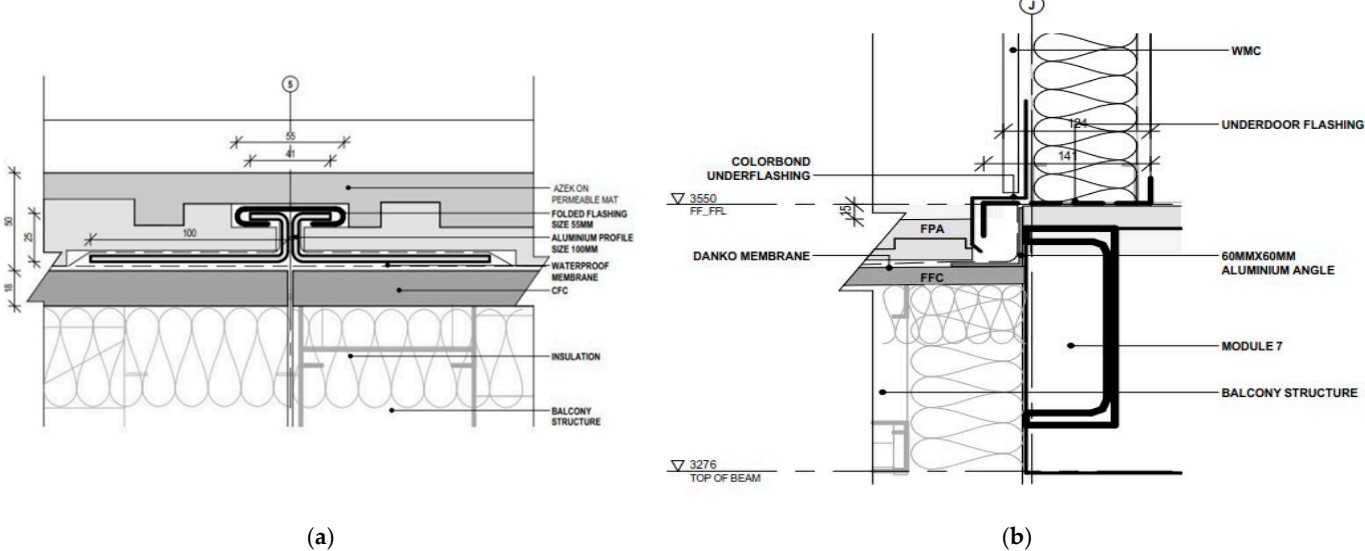

(a)                                           (b)

**Figure 7.** *Cont.*

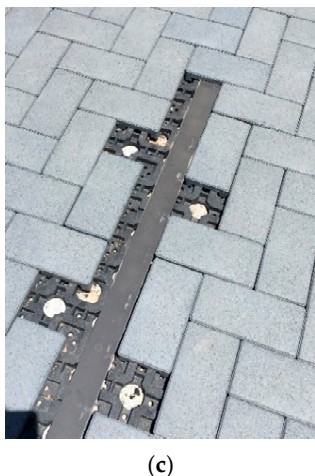
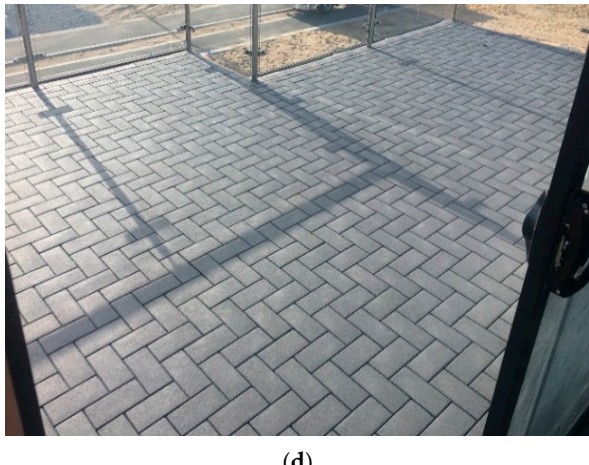

(**c**)    (**d**)

**Figure 7.** (**a**) Silo flashing running down the balcony pictured in (**c**); (**b**) overflashing design is waterproof, it aligns with the installation and disassembly process (top) and is concealed by a recycled rubber and plastic interlocked paving system, as in (**d**).

### 3.5.2. Plumbing System

The specification of plumbing systems is important since, typically, these services are not disassemblable. The pipework selected for L3 was a disassemblable plumbing system made from PEX piping with brass connection fittings. The brass fittings have specially designed teeth that bite into the piping, securing the pipework in position; however, these fittings can be disengaged using a specific tool, which allows removing and refitting the pipe. The plumbing system was designed to enable the isolation of modules and disconnection prior to the building being relocated. This aim was achieved through locating ceiling access panels in the bathroom module. From the cavity in the roof space of the bathroom modules, the potable and waste systems can be disconnected during disassembly or deconstruction.

### 4. Discussion

A defined theoretical gap exists in the current literature, namely, the reuse of building components [28], design for disassembly [14] and the circular economy in practice [13]. This study documents the identified design criteria, which facilitates an extension of the literature by addressing the three aforementioned research gaps. It details the integration of practice and theory by adopting the case study method and adds an empirical study to a theory based topic. It describes each of the connection details designed and implemented in the case study in an effort to divert the flow of building materials from waste streams and promote the reuse of second life products. This design research has produced a resilient, adaptable case study which, owing to its design, managed to incorporate 76% of reusable material and 10% of recyclable material and to achieve a complete disassembly and relocation while creating only <1% of waste in the process.

Starting with eight redundant structural steel framed modules, the first research question was addressed, as this building was designed around the waste resources available, thus saving 16.5 tonnes of steel. The steel frame structures were redesigned to facilitate an appropriate space and each building element was assessed, with the level of disassembly critiqued as barriers. After the barriers of each element were identified, novel solutions were implemented to enable the disassembly of the majority of the components, answering the second research question. The use of standardized material sizing and the introduction of waste material into the building enabled the building to be designed in line with the circular economy theory. Working with an experienced modular building company allowed the research team to investigate alternatives that would promote the circular economy in the building industry against the standard industry practice.

## 5. Conclusions

Typically, lightweight steel structured prefabricated buildings enable substantial material saving over traditional construction typologies, such as concrete and brick buildings [18,42,57]. Designing within parameters that allow parts of a building to be transported considerably increases the likelihood that they will not have to be demolished [58]. If the building structure is not required for a second life project, it has the ability to be stored until end of life. Storing it will allow it to be moved to an appropriate facility where material separation can be conducted in an organized, controlled facility, leading to increased material recovery. Most building sites have limited room for onsite waste sorting, and buildings connections are predominantly chemically bonded and often require extreme force to remove, leading to the damage of the components themselves. The difficulty of disconnection is compounded on including the variables associated with onsite construction, such as inclement weather, the lack of education and high labor rates. Therefore, demolition is still favored as the preferred method over deconstruction and disassembly.

Hence, in this study, through multiple design iterations, each detail of the building was considered. The goal was to achieve an aesthetic that reflected standard construction practice, yet offered the advantage of also being disassemblable. This aim led to the investigation and connection of each material used within the case study. Every effort was made to ensure that the building could be relocated as a complete building, in eight modules. Removable internal linings were introduced, should renovation be necessary to prolong the life of the building in location, aiming at material saving. To address the third research question, disconnection points are shown to be essential in furthering circular economy in the construction industry, and L3 has presented a high quality alternative to challenge the status quo in building. It is hoped that this example will challenge the negative perception regarding prefabricated construction in Australia as well as the values of potential buyers and occupants when the material savings and environmental benefit of this construction typology are presented.

## 6. Limitations and Future Research

The first limitation relates to building information modelling (BIM). BIM has the features to enable the development and uptake of circular economy in the construction industry. The use of advanced BIM software enables accurate documentation of the materials used in a construction project, allowing these to be tracked and stored across the building's life cycle, before finally becoming a materials inventory at end of life. This material inventory allows the mining of decommissioned buildings for future resources. However, a limitation, both nationally and internationally, is the lack of an open access platform that publishes these material banks, meaning these valuable components cannot be advertised and sold on. The design process needs access to component dimensions and decommissioning dates to introduce second life components and to, thus, close the loop in the material flow of construction components.

The second limitation is based on a practical learning concerning the application of circular economy that would benefit the building industry. The magnetic fixed ceiling panels were so easy to take off that, after several removals, the steel backing channels were damaged, and the magnets often filled with swarf, creating a suboptimal bond. The spacing of the metal backing was also moved to the external edges of the sheets (originally inset by 130 mm by the supplier) to create a flatter ceiling finish. Future research should endeavor to work closely with industry partners and have greater time for subcontractor participation in research projects. Subcontractors priced this project as standard installation and had no margin for time spent upskilling workers on new techniques.

Third, there was a notable design flaw in the roof because the roof battens were spaced at their maximum span. This resulted in multiple kinks in the roof sheets when riggers, roof plumbers and solar panel installers worked on the building. The overflashing design, which is responsible for waterproofing the connection of the color bond steel and timber wall cladding, worked in theory. However, the flashing had to be removed for the

transportation of the building, which affected the practicality of the design, given that it resulted in increased onsite work. Flashing design should be an ongoing consideration for research.

Last, to date, electrical connections are limited in their disconnectibility, and this should remain an area of focus for future research. The significant danger associated with unlicensed electrical work has led to a reluctance to develop a plug connection system through electrical circuits. The research team does not wish to see the work of licensed electrical contractors removed, but rather, wishes to see a system created that can be easily adapted without the need to hard wire each circuit. In this regard, connection systems used for automotive, or soft, furnishing, such as desk plugs, could carry a suitable number of apertures needed for most domestic housing situations.

**Author Contributions:** Conceptualization, T.M.O. and R.M.; methodology, T.M.O. and R.M.; formal analysis, T.M.O.; investigation, T.M.O.; resources, G.M.M.; data curation, T.M.O. and R.M.; writing—original draft preparation, T.M.O. and R.M.; writing—review and editing, T.M.O. and R.M.; supervision, G.M.M. and H.-Y.C.; funding acquisition, G.M.M. All authors have read and agreed to the published version of the manuscript.

**Funding:** This research was funded by the ARC ITTC–CAMP.H (Centre for Advanced Manufacture of Prefabricated Housing) that is thanked for funding through grant number IC150 1000 23. Zimi Technology has partnered with Curtin University in a state awarded PhD Fellowship research project J0546/201801.

**Institutional Review Board Statement:** Not applicable.

**Informed Consent Statement:** Not applicable.

**Acknowledgments:** The authors would like to acknowledge Fleetwood Building Solutions for their support throughout the construction and delivery of this project.

**Conflicts of Interest:** The authors declare no conflict of interest.

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
