# Peer review of "Interconnections: An Analysis of Disassemblable Building Connection Systems towards a Circular Economy"

_buildings, doi:10.3390/buildings11110535_

Round 1
Reviewer 1 Report
This is a very sound paper that covers an important issue. The case study is well presented and the methodology and research techniques well substantiated and explained. The findings are sound and transferable. This is a good addition to the debates and ideas around the circular economy in building. The background and theoretical foundations could be expanded a little to include more literature background on similar explorations; especially how this project shows the real application of other theoretical research.
There are a couple of minor corrections required: Figure 5 seems to have the images the wrong way around? Table 2 seems to be the same as Table 1?
Reviewer 2 Report
- The term “a circular economy building” is grammatically incorrect (unless this is a proper title, which does not appear to be in its current form). Either revise, or provide a concise definition and reference for what this term is.
- In describing the three connection types in the abstract, “waterproof” is technically a “non-structural” connection. As such, the authors are urged to either adopt 2 connection categories, or adopt an alternative term for “non-structural”.
- In the introduction, the description of prefabricated buildings (which focuses primarily on why “high quality” prefabricated buildings are required) does not aptly setup the importance for the three research questions. The introduction should describe specifically why research addressing these three questions is needed.
- The 3R’s hierarchy should be introduced and explained in Section 2.
- In describing the process for the design and construction of the case study, was the research team the only stakeholders involved? If so, do they have the appropriate engineering and design accreditation? It would be helpful to provide details of the other stakeholders involved in the project.
- Figure 3 -> how can something be infinitely reusable? This is misleading. Upon what basis is made for ensuring the timber/steel are only reusable 3 times?
- Should the word “join” be “joint”? “Join” is a verb, whereas “joint” is a noun.
- One thing missing in the case study is justification for why certain decisions are made. For instance, the comparison of the micropile system vs. the traditional footing design: it would be helpful to provide an LCA (life cycle assessment) justification for this decision rather than saying the micropile system can be reused.
- Overall, this paper presents information for the case study in the form of a design report, rather than a scientific paper. There needs to be a way to ascertain or evaluate the connection details rather than simply stating the connections used can facilitate disassembly. No comparison to alternatives exists, which is a current flaw.
- In the discussion, “cost” is brought up as being a theoretical gap, and yet this is not established in the initial research questions.
- The first sentence in the conclusion needs further explanation. In many cases, more materials are required for a prefabricated building on a strict comparison basis (up to 15% more materials according to: Kamali, K. Hewage, A.S. Milani, Life cycle sustainability performance assessment framework for residential modular buildings: Aggregated sustainability indices, Build. Environ. 138 (2018) 21-41.)
